# SWAR: A Deep Multi-Model Ensemble Forecast Method with Spatial Grid and 2-D Time Structure Adaptability for Sea Level Pressure

**Jingyun Zhang [1], Lingyu Xu [1,\*] and Baogang Jin [2,\*]**

1   Department of Computer Engineering and Science, Shanghai University, Shanghai 200444, China
2   Beijing Institute of Applied Meteorology, Beijing 100029, China
\*   Correspondence: xly@shu.edu.cn (L.X.); jinbaogang2021@163.com (B.J.)

**Abstract:** The multi-model ensemble (MME) forecast for meteorological elements has been proved many times to be more skillful than the single model. It improves the forecast quality by integrating multiple sets of numerical forecast results with different spatial-temporal characteristics. Currently, the main numerical forecast results present a grid structure formed by longitude and latitude lines in space and a special two-dimensional time structure in time, namely the initial time and the lead time, compared with the traditional one-dimensional time. These characteristics mean that many MME methods have limitations in further improving forecast quality. Focusing on this problem, we propose a deep MME forecast method that suits the special structure. At spatial level, our model uses window self-attention and shifted window attention to aggregate information. At temporal level, we propose a recurrent like neural network with rolling structure (Roll-RLNN) which is more suitable for two-dimensional time structure that widely exists in the institutions of numerical weather prediction (NWP) with running service. In this paper, we test the MME forecast for sea level pressure as the forecast characteristics of the essential meteorological element vary clearly across institutions, and the results show that our model structure is effective and can make significant forecast improvements.

**Keywords:** MME; NWP; forecast; sea level pressure; deep learning

## 1. Introduction

The concept of NWP was put forward more than 100 years ago, and was first put into operation in the 1970s. The NWP depends on the numerical model of the forecast institution, which produces the forecast results by running its numerical model. With the development of society, the accuracy of forecasting is more and more important as the forecast results affect the decision-making of various industries. The earliest NWP used the deterministic prediction method. However, due to the uncertainty of atmospheric motion, ensemble forecast was proposed and developed to replace the deterministic method. Ensemble forecast is based on uncertainty, and it applies different disturbances to the initial condition of the numerical model to produce multiple forecast results. Each forecast result generates an ensemble member. These forecast results are presented in the form of probability, or aggregated to a certain one, so as to improve the accuracy and robustness of the forecast [1]. However, the limitations of the single numerical model itself, for example, making it difficult to get significant forecast improvement by optimizing initial field, dynamic model, physical parameterization, etc. [2], which will affect the forecast skill, shows that it is necessary to adopt the MME forecast as different numerical models from their institutions have different forecast characteristics, and by integrating them the forecast quality can get a further improvement. Hagedorn et al. discussed the reasons for the success of MME [3] in 2005, and Weigel et al. systematically proved that even if a single model integrates a model whose forecast skills are inferior to its own, it is still possible to obtain a better forecast model than on its own [4]. A key aspect of making a successful MME forecast is to integrate

the spatial-temporal features of different models, and establish the mapping relationship to the ground truth. However, the forecast results of the numerical model have a special spatial-temporal structure. The forecast results issued by many existing institutions present a grid structure [5] formed by the intersection of longitude and latitude lines in space. This structure can be regarded as an image problem and each point on the grid is equivalent to a pixel on the image. In time, compared with the traditional one-dimensional time, it has a special two-dimensional time structure, namely the initial time and the lead time. The initial time is the time when the forecast institution runs its numerical model and the lead time is based on the initial time and make an offset or shift to the future. In fact, most of forecast institutions belong to this mode as they need to maintain continuous forecasting. Each initial time starting a new run will obtain a series of new forecast results about the future which refer to the lead time, thus presenting a rolling forecast mode that many forecast results will overlap for a period of time. These characteristics cause many ensemble methods to have limitations in further improving forecast quality. Therefore, we need an MME forecast model that can make full use of the special spatial-temporal structure. For sea level pressure, the spatial-temporal characteristics of forecasting vary clearly across institutions, which provide a foundation for the MME forecast model to make forecast improvements. Sea level pressure is important; it directly or indirectly affects many meteorological elements, such as air temperature, wind, etc., and is also closely related to the formation of some extreme weather. Therefore, it is of great significance to improve the forecast quality of sea level pressure.

## 2. Related Work

Typical statistics based ensemble methods include bias-removed ensemble mean (BREM), superensemble (SUP) [6], multiple linear regression based superensemble (LR-SUP) [7], quantile regression (QR) [8], and bayesian model aeraging (BMA) [9], etc. Some scholars also applied machine learning-based methods. For example, support vector machines (SVM) and random forests (RF) are used for MME forecast in many fields [10–12]. In the field of meteorology, Bin Wang et al. used SVM and RF to integrate the temperature and precipitation in CMIP5 global climate model and experimental results show that the integration method performs better than EM and BMA in each month [13]. Taking the surface temperature (SAT) in China as the analysis, Yuming Tao et al. integrated 24 general circulation models (GCM) using the residual based clustering tree and the results show that the root mean square error (RMSE) of multi-model ensemble model (MMEM) is lower than the optimal GCM model and the ensemble mean of these models in each season [14]. With the intensification of climate change and the requirement of further improving forecast accuracy, the limitations of traditional methods are gradually highlighted, and the combination with deep learning becomes key to making significant improvement. As a non parametric model, the neural network does not make any assumptions about the data distribution. It uses infinite dimensional parameters to describe the data, and it proves that the neural network can theoretically fit any function by universal approximation theorem. In this regard, Scher [15] studied the output of the GCM model through the neural network, proving that the trained neural network has the potential to achieve a purely data-driven forecast. Kumar et al. applied artificial neural network (ANN) to MME forecast [16] and analyzed monsoon precipitation in India. The experimental results showed that ANN has higher forecast skills than GCM models and their ensemble mean. There are also scholars that combine traditional MME methods with deep learning to serve other prediction tasks, such as the work of jiwon Kim et al. [17]. They used BMA to integrate the surface temperature, surface humidity, and other cofactors of multiple regional climate models (RCM), using multiple linear regression and deep neural network (DNN) to serve the downstream tasks about sea ice concentration prediction. Focusing on the problem that BMA is sensitive to the selected members of the ensemble, Dayang Li et al. proposed a recurrent neural network based on variational Bayes (VB-LSTM) [18] to keep accuracy and robustness while making the MME forecast. Peter Grönquist et al. [19] adopted U-Net

and multi-level residual networks with several ensemble members of global numerical model, and the results showed significant improvements in temperature and precipitation at 850 hPa. Yi-Fan Hu et al. [20] used the U-Net with attention mechanism to correct the bias of precipitation for global numerical model, and their model showed good performance in reducing RMSE and improving the threat scores while utilizing multiple auxiliary factors. In fact, most of current MME forecast methods did not focus on the special structure of numeric forecast results, and showed limitations in improving multiple dimensions of the original forecast simultaneously. Therefore, we propose a deep method for MME forecast that adopts shifted window attention and Roll-RLNN (SWAR). In our experiments, SWAR achieves best performance in nearly all indicators compared with other methods. In addition, SWAR only needs several forecast results of different numeric models without any additional meteorological factors, so it is also suitable for the environment with insufficient forecast materials.

## 3. Method

From the perspective of information fusion, MME forecast belongs to the decision level fusion. For the neural network, the task to integrate the special spatial-temporal features of different models can be converted to utilize and aggregate the information of multiple dimensions from different sources, and make a final decision based on them. In the following sections, we first introduce the feature extraction method at the spatial level, then the special temporal level with 2-D time structure, and finally incorporate them into an overall framework to achieve deep MME forecast.

### 3.1. Window Self-Attention

Attention mechanism is an important concept of deep learning. It has been widely used in many places, including recommendation [21], prediction [22,23], speech recognition [24], and image segmentation [25,26], etc. The self attention mechanism can spontaneously focus on the relationship between elements in a set, and, based on this, the neural network can extract deeper features. However, the self attention mechanism focuses on the global relationship of all elements, that is, each element will participate in relationship calculation and weighted aggregation with all other elements in the set. In fact, this method will cause a quadratic increasing cost of computation and one element in the set may not have strong relations with all other elements. Therefore, the global relationship computing may also introduce noise. At the same time, meteorological elements relate locally and spatially as atmosphere is a continuous motion system. So, for the above problems and characteristics, we introduced the window self-attention mechanism. Window self-attention had some great applications before [27,28]. It divides the original area $W$ with size $r \times c$ into several sub-areas of the same size, each area is marked as a sub-window, and elements inside the sub-window will be fused by the self-attention mechanism. For a sub-window with size $r_s \times c_s$, where $r = t_n \times r_s$, $c = t_m \times c_s$ and $t_n, t_m \in \mathbb{N}_+$, the division way is shown on the left of Figure 1. We mark the hidden representation in the sub-window $k$ as $\mathbf{H}_k^W$, the data after attention calculation as $\mathbf{Z}_k^W$, where $\mathbf{H}_k^W \in \mathbb{R}^{r_s \times c_s \times d_{\mathrm{rep}}}$, $\mathbf{Z}_k^W \in \mathbb{R}^{r_s \times c_s \times d_z}$ and $1 \leq k \leq t_n \times t_m$, and the operation of self-attention, window self-attention as SA, WSA respectively, and then the process can be expressed as follows:

$$
\begin{aligned}
\mathbf{Z}^W = \mathrm{WSA}\left(\mathbf{H}^W\right) &= \left\{ \mathrm{SA}\left(\mathbf{H}_k^W\right) \right\}_{k=1}^{t_n \times t_m} \\
&= \left\{ \mathrm{softmax}\left( \frac{\mathbf{Q}_k^W \left(\mathbf{K}_k^W\right)^T}{\sqrt{d_h}} \right) \mathbf{V}_k^W \right\}_{k=1}^{t_n \times t_m}
\end{aligned}
\tag{1}
$$

In Equation (1) $\mathbf{Q}_k^W, \mathbf{K}_k^W, \mathbf{V}_k^W$ are transformed from $\mathbf{H}_k^W$ by applying multiplication with $\mathbf{W}_a, \mathbf{W}_b, \mathbf{W}_c$ respectively where $\mathbf{W}_a, \mathbf{W}_b \in \mathbb{R}^{d_{\mathrm{rep}} \times d_h}$, $\mathbf{W}_c \in \mathbb{R}^{d_{\mathrm{rep}} \times d_z}$ and $d_h$ is the dimension of $\mathbf{Q}_k^W$ and $\mathbf{K}_k^W$.

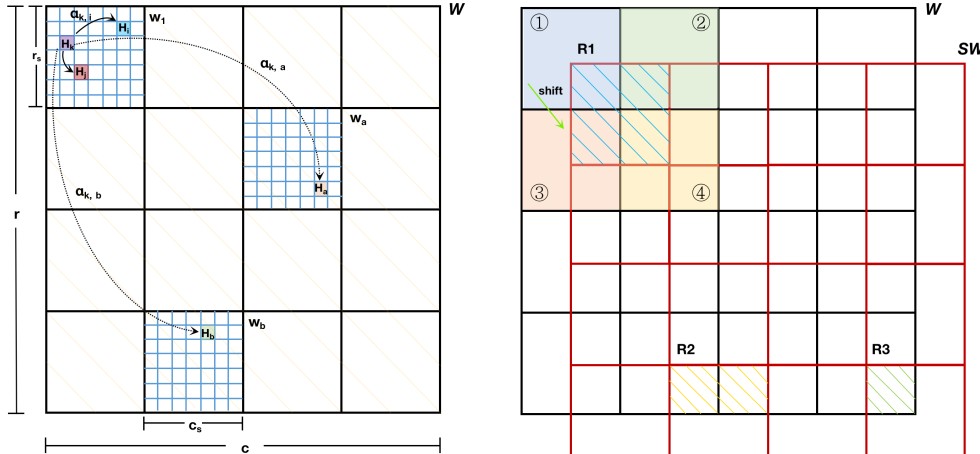

**Figure 1.** On the left side, window self-attention first divides the original window $W$ with size $r \times c$ into several sub-windows of the same size $r_s \times c_s$. Each sub-window contains elements formed by a grid structure, as shown by the blue grid line, and then window self-attention will fuse the information in each sub-window. $\alpha_{x,y}$ represents the influence on grid point $y$ by grid point $x$, as shown by the arrows. On the right side, the hatching parts in shifted window $SW$ represent the aggregation process from different sub-windows. For example, the hatching part R1 in shifted window $SW$ aggregates the information from sub-windows 1 to 4 in window $W$.

By the dividing window, we only need to focus on the attention calculation of each sub-window, and this method greatly reduces the computational complexity. At the same time, window self-attention fuses elements in the same sub-window and this is equivalent to introducing the locality of convolution process in convolutional neural network (CNN) which suits the characteristic that meteorological elements are related locally and spatially. For example, the element $\mathbf{H}_k$ may relate with $\mathbf{H}_i, \mathbf{H}_j$ in the same sub-window $W_1$, as shown in the left of Figure 1. In fact, the relations of meteorological elements exist not only in the sub-window, but also between sub-windows. For example, the element $\mathbf{H}_k$ in sub-window $W_1$ may relate with $\mathbf{H}_a$ in sub-window $W_a$ and $\mathbf{H}_b$ in sub-window $W_b$. The method for this part will be discussed in Section 3.2.

### 3.2. Shifted Window Attention

The previous section mentioned that there are also relations between the sub-windows, so in this subsection we use shifted window attention to make information interact between sub-windows. On the right side of Figure 1, we apply the right down shifting to all sub-windows in the global window $W$, which can be decomposed into $\lfloor \frac{c_s}{2} \rfloor$ steps of right shifting in the horizontal direction, and then $\lfloor \frac{r_s}{2} \rfloor$ steps of down shifting in the vertical direction. We mark the global window after shifting as $SW$.

$$SW_{(i,j)} \leftarrow \begin{cases} W_{(i,j)} \\ W_{(i,j+1)} \\ W_{(i+1,j)} \\ W_{(i+1,j+1)} \end{cases} \tag{2}$$

The arrow in Equation (2) represents the aggregation relationship. The shift sub-window $SW_{(i,j)}$ aggregates the information of four different sub-windows $W_{(i,j)}$ to $W_{(i+1,j+1)}$ at a higher level, where $(1,1) \leq (i,j) \leq (t_n - 1, t_m - 1)$. After shifting, the next time calculating window self-attention will have the same aggregation relationship, as shown by Equation (3).

$$W_{(i+1,j+1)} \leftarrow \begin{cases} SW_{(i,j)} \\ SW_{(i,j+1)} \\ SW_{(i+1,j)} \\ SW_{(i+1,j+1)} \end{cases} \tag{3}$$

By cross-iterating Equations (2) and (3), all the sub-windows can have information interactions at higher levels and it can help the neural network to fuse more features of different scales. For the boundary cases of the operation above, for example, in Equation (2) the number of sub-windows is less than four when $i = t_n$ or $j = t_m$, as shown by the hatching area R2 and R3 in the right of Figure 1. For these cases we need additional networks to calculate the attentions with window size $\lfloor \frac{r_s}{2} \rfloor \times r_c$ and $\lfloor \frac{r_s}{2} \rfloor \times \lfloor \frac{r_c}{2} \rfloor$. However, this method will introduce attention networks with different scales, which cannot perform batch calculation at the same time and will cause low speed of model training. For this problem, [29] proposed a cyclic shifting method, which achieves unified batch computation among hatching areas R1, R2 and R3.

*3.3. Roll-RLNN for 2D Time Structure*

In the traditional time series problems, the time structure is one-dimensional, which means that the RLNN based on this structure can only transmit information in a single direction. However, in a numerical forecast, the time structure is two-dimensional, and the initial time and the lead time jointly present a rolling forecast mode that we mentioned in Section 1. The neural network information transmitted along either of the two dimensions will ignore the temporality of the other, and this means the raw RLNN cannot effectively utilize the features of 2D time structure. Notice that the temporality here focuses more on having the same temporal orientation and temporal interval between each element within a collection of sequences. To further discuss the problem, we define two sequential collections with same time interval $d$, the collection of initial time $\mathbf{T} = \{T_k\}_{k=1}^n$ and lead time $\mathbf{L} = \{L_v\}_{v=1}^m$ and we mark the forecast result with $T_k$ as the baseline or start time and $L_v$ as the offset to the future as $\mathbf{X}_{+L_m}^{T_1}$. Then, for the rolling forecast mode, the $k$th forecast will output a series of forecast results about future referred to the lead time which can be expressed as $\{\mathbf{X}_{+L_1}^{T_k}, \mathbf{X}_{+L_2}^{T_k}, ..., \mathbf{X}_{+L_m}^{T_k}\}$. However, after the forecast $\mathbf{X}_{+L_m}^{T_k}$ of max lead time, the next forecast will be rolled back to $\mathbf{X}_{+L_1}^{T_{k+1}}$ and the change from the time $(T_k, L_m)$ to $(T_{k+1}, L_1)$ will break the previous temporality that is necessary for RLNN as it will change the temporal orientation and interval. Therefore, most deep methods do not learn by directly flattening and connecting the two temporal dimensions, but learn along one of the dimensions while different values in the other dimension are set as different batches of training samples that are independent of each other. The limitations of learning along one single temporal dimension have been mentioned before. For example, learning along the lead time through RLNN always starts with a fixed temporal state $\mathbf{s}_{\text{init}}^{T_k}$, or $\mathbf{s}_{\text{init}}^{T_k}$ can be divided into more temporal states like $\mathbf{h}_{\text{init}}^{T_k}$ and $\mathbf{c}_{\text{init}}^{T_k}$ in LSTM. These fixed states are usually set before training and the fixed feature, which means that the forecast results based on different initial times are independent from each other. In fact, forecast results from different initial times are not independent as these forecast results produced by their numeric model follow the same rules, but there are slight differences in initial condition. To clearly discuss this, we present the transform relationship between 1-D and 2-D time in numeric forecast as shown by Equations (4) and (5).

$$t_{\text{std}}(T_k, L_v) = T_k + L_v \tag{4}$$

$$t_{\text{std}}(T_k, L_v) = t_{\text{std}}(T_k - \alpha d, L_v + \alpha d) \tag{5}$$

where $\alpha \in \mathbb{Z}$ and $t_{\text{std}}$ is defined as the standard time to describe 1-D time after the transform of 2-D time. $d$ is the time interval for initial time and lead time. These equations imply that for a certain standard time $t_\sigma$, there are multiple forecast results with different initial time

and lead time to describe the atmospheric state, and these forecast results follow the same rules and are produced to fit the ground truth as close as possible, which means that, for a numeric model with acceptable confidence, if one forecast gives the result $\mathbf{X}_{t_\sigma}$, then the other forecast results will turn to give the results that are close to $\mathbf{X}_{t_\sigma}$. Therefore, forecast results from different initial times are not independent, which is not suitable for the fixed temporal states that will ignore the temporality between different initial times. To cause information interaction between different initial times, we need to find a more suitable temporal state to replace the fixed one. We mark $\mathbf{X}_{+L_1}^{T_k}$ as the forecast result at standard time $t_\sigma$, and $\mathbf{X}_{+L_1}^{T_k}$ has no temporal context as is the first lead when the temporal state $\mathbf{s}_{\text{init}}^{T_k}$ is fixed. To keep the temporality with the time interval $d$ or provide the temporal context for $\mathbf{X}_{+L_1}^{T_k}$, we need to choose a forecast result at time $t_\sigma - d$. From Equations (4) and (5), we can derive the Equation (6).

$$
\begin{aligned}
\mathbf{F}_{t_\sigma - d} &= \left\{ \mathbf{X}_{+L_j}^{T_i} \mid T_i + L_j = t_\sigma - d \right\} \\
&= \left\{ \mathbf{X}_{+\beta d}^{T_k - \beta d} \mid \beta \in \mathbb{N}_+ \right\}
\end{aligned}
\tag{6}
$$

where $\mathbf{F}_{t_\sigma - d}$ represents a series of forecast results at standard time $t_\sigma - d$. For multiple candidate results, we need to select the most reliable one that can serve as the basis for the neural network to learn along both dimensions. Notice that the forecasting skills of the numerical models of each institution decrease with the lead time going forward [30], which is the characteristic of NWP. When $\beta = 1$, the numerical models have the optimal forecasting skills, and the forecast of the first lead can be used as the basis of subsequent forecasts along two dimensions. For example, $\mathbf{X}_{+L_1}^{T_k}$ can underlie both $\mathbf{X}_{+L_2}^{T_k}$ and $\mathbf{X}_{+L_1}^{T_{k+1}}$ as they have the same temporal orientation and temporal interval, which are important for an RLNN to learn from the sequential data. So, by substituting $\beta = 1$ into Equation (6) and considering Equations (4) to (5), we get the forecast result $\mathbf{X}_{+L_1}^{T_{k-1}}$. Based on this, we propose the structure of Roll-RLNN as shown in Figure 2.

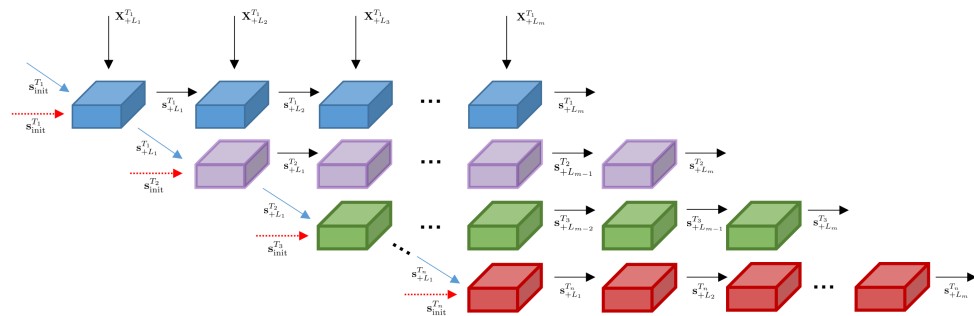

**Figure 2.** Structure of Roll-RLNN for 2-D time in numeric forecast is shown. The same colors represent the RLNN units which process the leading forecasts started at the same initial time, while the different colors represent the different initial times.

Where different red dotted arrows indicate that the structure based on traditional RLNN uses fixed temporal states as the initial input of the network at starting stage of their corresponding initial time. After input, the temporal states are transmitted along the lead time, i.e., the horizontal direction of the black arrow in Figure 2. The Roll-RLNN replaces the red dotted arrows with blue solid arrows that represent a new form of transmission by which information interaction of the 2-D time structure can be better achieved for numeric forecast.

### 3.4. Model Structure

In the previous section, we discussed the feature extraction methods at the spatial level and the temporal level with 2-D time structure. In this section, we propose the structure

of the SWAR Model (Shifted Window Attention and Roll RNN Model), which is used to aggregate the features of different levels and make MME forecast to improve the forecast quality.

In this model, the dimension of the forecast institution or numerical model is introduced. For multiple sets of forecasting results $\mathbf{X}_{T_k,L_v}^{(r)} = \{\mathbf{X}_{i+L_v}^{T_k}\}_{i=1}^r$, $r$ is the number of forecast institutions or numeric models, as shown in part 1 of Figure 3. $\mathbf{O}_{T_k,L_v}^{(1)}$ represents the final output of MME forecast at 2-D time $(T_k, L_v)$ by using $\mathbf{X}_{T_k,L_v}^{(r)}$ as input, and for an effective model, $\mathbf{O}_{T_k,L_v}^{(1)}$ has a higher forecast quality than $\mathbf{X}_{T_k,L_v}^{(r)}$. Notice that our model combines multiple sets of forecast results with different spatial and temporal features, so the information to make the output $\mathbf{O}_{T_k,L_v}^{(1)}$ not only comes from the current forecast result $\mathbf{X}_{T_k,L_v}^{(r)}$, but also from the spatial-temporal context of the current forecast. Spatially, our model divides the whole area into several sub-windows to make self-attention according to the Section 3.1, and then apply the shifted window attention according to the Section 3.2 to aggregate features at a higher level from different sub-windows, such as part 2 of Figure 3, where the red sub-window shifts to the red dashed one, and all other sub-windows conform to the same way. For the integration of multiple forecast results, we consider different institutions or numeric models as the channel dimensions, which is a basic concept in computer vision. Thus, by applying the convolution process, we can aggregate features of forecast results from different institutions directly. Specifically, we extracted spatial features from the multiple sets of forecast results within each sub-window $w$ by applying convolution operations of kernel size $1 \times 1$ and $r_s \times c_s$ with $1 \leq w \leq r_s \times r_c$ and fuse the spatial features at different scales to obtain the aggregated feature $\mathbf{H}_{T_k,L_v}^w$ where for the dimensions of $\mathbf{H}_{T_k,L_v}^w$, OC is the output channel size of convolution, and $\text{rep}_h$ is the last hidden dimension. The ellipsis in prefix emphasizes that the dimension of the current feature which is displayed belongs to an overall dimension framework. For example, the dimensions that the ellipsis represents also include the initial time, lead time, sub-window index, etc. The aggregation methods are mainly expansion and combining. Specifically, the features after convolution with kernel size $r_s \times c_s$ were used as the global features of each sub-window by applying $r_s$ times and $c_s$ times expansion respectively along the line of longitude and latitude, and then aligns and combines with features output from $1 \times 1$ convolution, in which the combining process is along the last dimension of features. Then, the spatial feature $\mathbf{H}_{T_k,L_v}^w$ is incorporated into the context of the temporal state $\mathbf{S}_{T_k,L_v-1}^w$ as shown by the horizontal solid arrow with black color, or $\mathbf{S}_{T_k-1,L_1}^w$ when the current forecast is the first lead as shown by the solid blue arrow with an asterisk, which is the process $F$ in Part 3 of Figure 3 and is also implemented by expanding and combining that are similar to the previous processes. Specifically, for $\mathbf{S}_{T_k,L_v-1}^w$ or $\mathbf{S}_{T_k-1,L_1}^w$, expand $r_s$ and $c_s$ times along the line of longitude and latitude respectively, and then flatten the last two dimensions of $\mathbf{H}_{T_k,L_v}^w$ into one dimension with the dimension size oc $\times$ $\text{rep}_h$, and finally align and combine with the spatial feature $\mathbf{H}_{T_k,L_v}^w$. The output of the process $F$ is successively passed through the window self-attention network and the shifted window attention network, which are marked as W-Attn and SW-Attn to obtain the feature $\mathbf{H}_{T_k,L_v}^{'w}$ which is locally and globally aggregated at a spatial level. To improve the adaptability of our model, we use residual networks as an aid in the above processes. At the same time, to make the aggregated spatial feature integrated with 2-D time structure, we apply a convolution process with kernel size $r_s \times c_s$ to $\mathbf{H}_{T_k,L_v}^{'w}$, which will output a spatial feature that can be used as the input of W-RLNN (Window RLNN) with $\mathbf{S}_{T_k,L_v-1}^w$ or $\mathbf{S}_{T_k-1,L_1}^w$ together. Compared with RLNN, W-RLNN does not directly learn the temporality from the global region $r \times c$, but learns from different sub-windows with each size $r_s \times c_s$ in batches. The benefits of this approach are similar to the window division in Section 3.1. The W-RLNN then outputs the updated temporal state for each sub-window as input to the next 2-D time. The transforming process near the W-LRNN in the part 3 of Figure 3 represents a series operations such as dimension exchange, reshaping, etc., to make the temporal states of 2-D time structure adapt to the

input of W-RLNN or the reverse process from the output of W-RLNN to the temporal states. At the same time, notice that the window partition size of W-RLNN is consistent with the partition size in Section 3.1. Therefore, by using shifted window attention, the temporal states from different sub-windows can also have information interaction at a higher level. RLNN has many implementations and we use LSTM as the basic unit in our model for the temporal part. Therefore, the temporal state $\mathbf{S}^{w}_{T_k,L_{v-1}}$ or $\mathbf{S}^{w}_{T_k-1,L_1}$ is divided into two parts, representing the long and short temporal state respectively, which are indicated by different colors. Through these processes, our model can better aggregate the features from different institutions that have different spatial-temporal forecast characteristics.

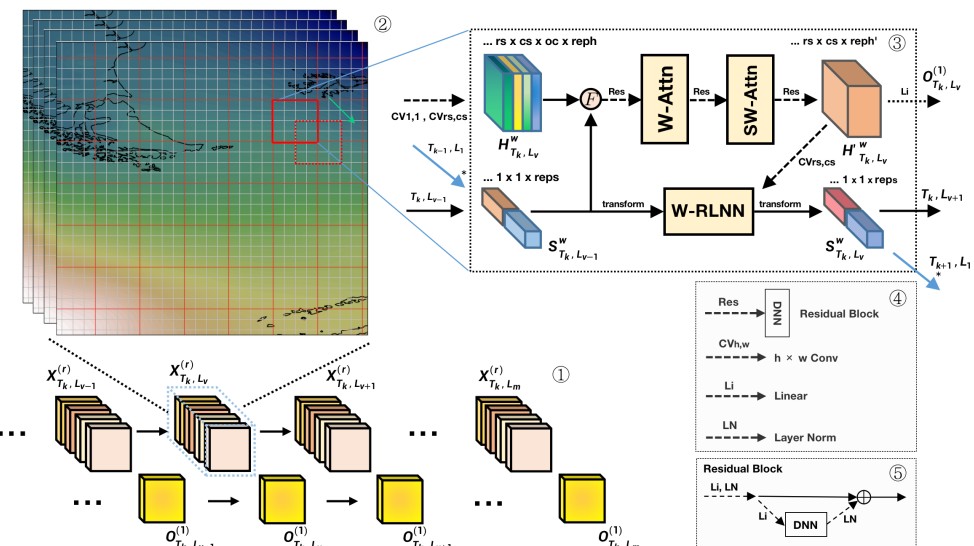

**Figure 3.** Structure of SWAR is shown. Parts 1 to 5 represent the input and output of the model, the forecast results of different institutions, the specific information aggregation process of the model, and the interpretations of some processes in part 3.

## 4. Experiment

### 4.1. Data Selection

The data of sea level pressure are derived from THORPEX Interactive Grand Global Ensemble (TIGGE) [31] that is integrated by European Centre for Medium-Range Weather Forecasts (ECMWF). The dataset collects numerical model forecast data from different forecast institutions, including ECMWF itself, National Center for Environmental Prediction (NCEP), United Kingdom Met Office (UKMO), Japanese Meteorological Administration (JMA), China Meteorological Administration (CMA), etc. Numeric models of different forecast institutions differ in the initial field, dynamical framework, physical parameterization, etc., leading to the different forecast skills represented at spatial and temporal level, which makes it possible for the MME forecast model to improve the forecast quality [3]. Forecast types usually include perturbed forecast and control forecast, where perturbed forecast is started by applying slight disturbances to the initial field which aim to simulate and control the uncertainty of atmospheric motion while control forecast is started by an analyzed initial field which usually has a higher forecast skill that the perturbed one. Therefore, for the forecast type we choose the control forecast. For the time parameters, these forecast data have the same time interval, which is 24 h for both two dimensions with the lead time from 24 to 168 h, and the initial time at 12:00 each day. For spatial parameters, the grid spacing formed by line of latitude and longitude were all 1°. For numeric models, data from four forecast institutions, ECMWF, NCEP, CMA and JMA are chosen as the input of MME forecast sources. Observation data or reanalysis data are generally used as the labels for model training. However, meteorological stations are not distributed in a grid structure at spatial level, and there are fewer locations for observation on the ocean, which cannot be well aligned with the forecast results, which means that direct interpolation will introduce

significant error. Therefore, to avoid unnecessary error, we choose the reanalysis data as a training label in our experiment. Compared with ERA-Interim, ERA5 is the latest reanalysis data of ECMWF, and had been optimized in spatial resolution, data assimilation, physical parameterization, etc. Overall, ERA5 is more reliable in uncertainty estimation and has the quality that it is closer to the ground truth than ERA-Interim. For the parameters of ERA5, it has the spatial resolution with the interval of latitude and longitude 0.25 degrees, and the temporal resolution with 6 h interval from 0 to 18. To align in time, the value at 12:00 on each day is used. At the spatial level, the ERA5 with high resolution can be sampled alternately on the grid structure to align with the forecast results that have lower resolution.

*4.2. Data Analysis*

We calculated the overall RMSE of sea level pressure in global range, including the average RMSE of 24–168 h leading, and found that there is a high forecast error region from 32.0° N to 59.5° N and 155° E to 177.5° W in December 2019 as shown in Figure 4. The high forecast error exists in each institution.

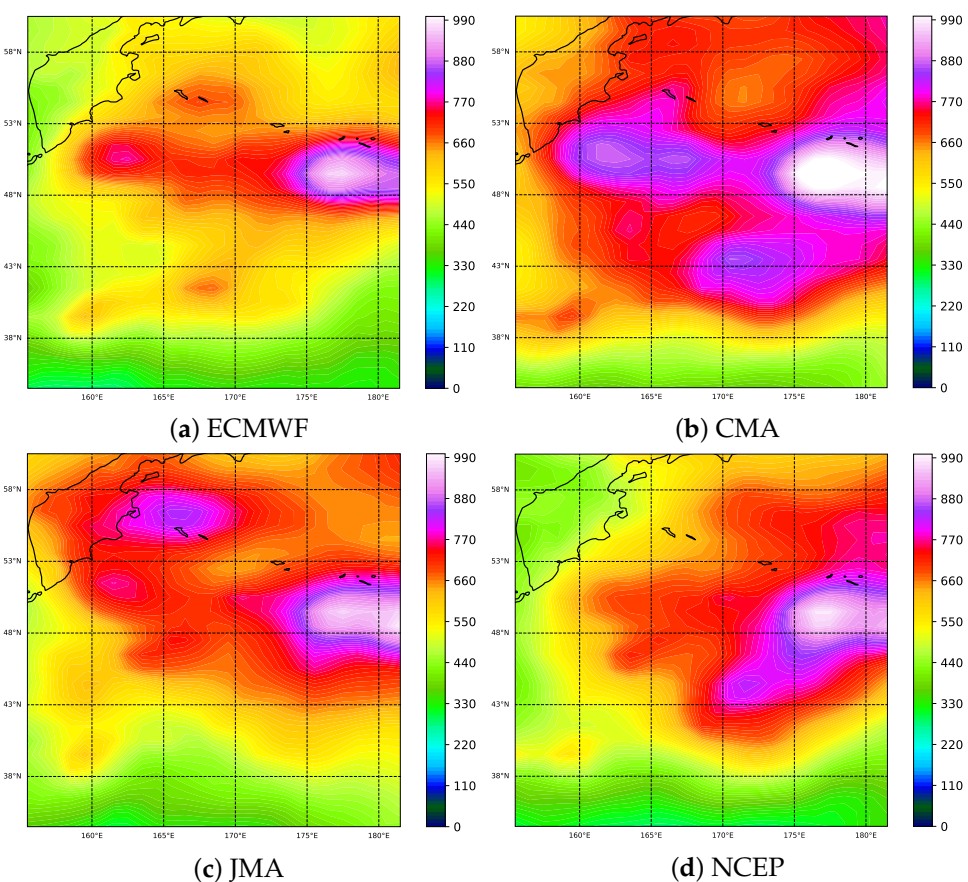

**Figure 4.** (**a**–**d**) show the RMSE distribution of forecast institutions for sea level pressure (Pa) at spatial level from 24 to 168 h lead time in December 2019.

Spatially, each institution presents a high error region centered at the latitudes from 47° N to 51° N and the longitudes from 175° E to 177.5° W with the forecast error decreasing radially. For the lead time, we calculated an average RMSE of 31 days for each lead time in December 2019 as shown in Figure 5. By following the same method, we calculate all the points in the region of Figure 4 where each point represents an element in the grid structure, and then calculate the average value of all the lead time for each point. Based on the calculation results of these points, we plotted the boxplots of RMSE for these institutions as shown in Figure 5. The results show that forecast errors increase with the lead time, which is one of intrinsic characteristics of NWP [30]. Among them, ECMWF has the best forecasting skill, especially in the short-term forecasts. However, as with other forecasting

agencies, the forecast error of sea-level pressure in the region increases rapidly with time ahead and fluctuates widely in the spatial extent, so it is important to improve the forecast quality in the region by means of MME forecasting.

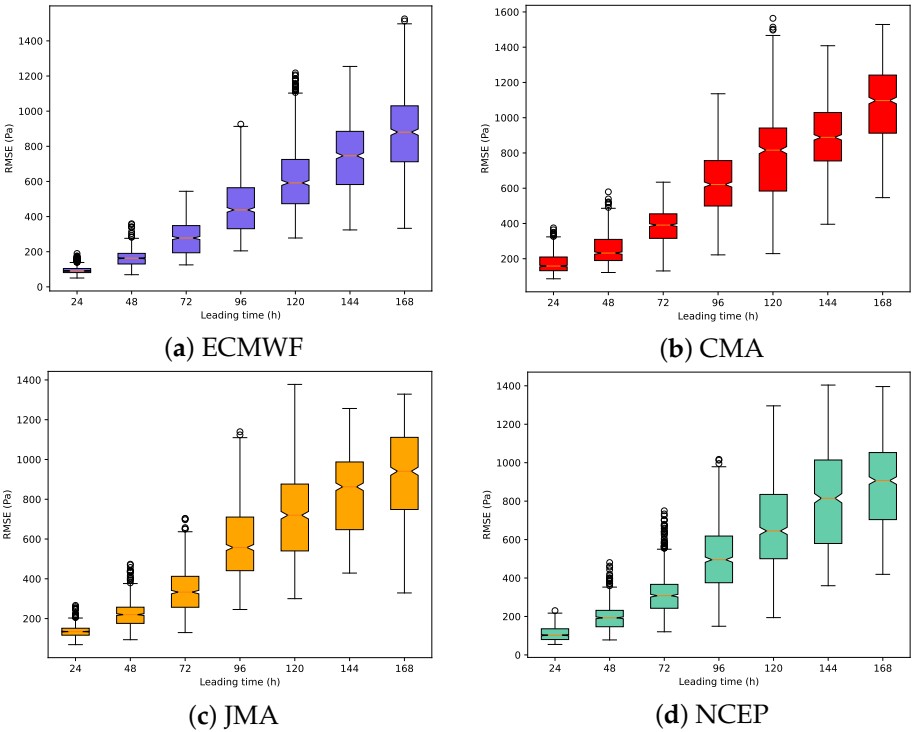

**Figure 5.** (**a**–**d**) show the RMSE distribution of forecast institutions for sea level pressure (Pa) in the lead time dimension from 24 to 168 h in December 2019 by the boxplots. The outliers in (**a**–**d**) indicate that each institution has spatial points with extremely high forecast errors.

*4.3. Metrics*

In this paper, three metrics, root mean square error (RMSE), mean absolute error (MAE), and pearson correlation coefficient (PCC) are used to evaluate the performance of MME forecasting models for sea level pressure. Due to the specificity of the two-dimensional time structure, we evaluate the integrated forecast effect separately for each lead time within December 2019 (24–168 h), so that we can assess the model effect more comprehensively. At the same time, to evaluate the performance at the spatial level, we give the distribution of the RMSE change rate in the experimental section. In the following, we give the calculation of the metrics $\{I^{(i)}_{+L_v}\}^3_{i=1}$ at the lead time $L_v$. We mark the forecast results to be evaluated as having forecasts, and each forecast is based on the current initial time with time steps ahead, as shown in Equations (7) to (9).

$$\mathbf{I}^{(1)}_{+L_v} = \text{MAE}_{(+L_v)} = \frac{\sum\limits_{k=1}^{n} |\mathbf{Y}^{T_k}_{+L_v} - \mathbf{O}^{T_k}_{+L_v}|}{n} \tag{7}$$

$$\mathbf{I}^{(2)}_{+L_v} = \text{RMSE}_{(+L_v)} = \sqrt{\frac{\sum\limits_{k=1}^{n} (\mathbf{Y}^{T_k}_{+L_v} - \mathbf{O}^{T_k}_{+L_v})^2}{n}} \tag{8}$$

$$\mathbf{I}^{(3)}_{+L_v} = \text{PCC}_{(+L_v)} = \frac{\sum\limits_{k=1}^{n} (\mathbf{Y}^{T_k}_{+L_v} - \overline{\mathbf{Y}}^{T_k}_{+L_v})(\mathbf{O}^{T_k}_{+L_v} - \overline{\mathbf{O}}^{T_k}_{+L_v})}{\sqrt{\sum\limits_{k=1}^{n} (\mathbf{Y}^{T_k}_{+L_v} - \overline{\mathbf{Y}}^{T_k}_{+L_v})^2}\sqrt{\sum\limits_{k=1}^{n} (\mathbf{O}^{T_k}_{+L_v} - \overline{\mathbf{O}}^{T_k}_{+L_v})^2}} \tag{9}$$

It should be noted that the multiplication operation in the above equations is the hadamard product. The above equations also include the calculation at spatial level, i.e., $\mathbf{I}_{+L_v}^{(i)} \in \mathbb{R}^{r \times c}$. Furthermore, the metrics in the temporal and spatial dimension in this paper can be expressed as Equation (10) and Equation (11), respectively.

$$I_{+L_v}^{(i)} = \frac{1}{r \times c} \sum_{p=1}^{r \times c} v_p, v_p \in \mathbf{I}_{+L_v}^{(i)} \tag{10}$$

$$\mathbf{I}^{(i)} = \frac{1}{m} \sum_{v=1}^{m} \mathbf{I}_{+L_v}^{(i)} \tag{11}$$

$I_{+L_v}^{(i)}$ indicates the overall metrics results at different lead times, while $\mathbf{I}^{(i)}$ indicates the overall results at each point on the grid space.

### 4.4. Model Selection and Main Hyper-Parameters

The background of the experimental data is described in Section 4.1, and the selected range is from 32.0° N to 59.5° N and 155° E to 177.5° W, forming a grid region with a latitude and longitude spacing of 1°, with a total of 784 spatial points. The time period from June 2019 to November 2019 was selected as the model training data, and December 2019 was selected as the test data as a way to verify the inference capability of the models. In order to fully validate the effect of model integration forecasts, we validate the lead time from 24 to 168 h respectively, and the specific metrics for validation are given in Section 4.3. We also choose some models based on statistics, machine learning, and deep learning as baseline. For machine leaning, we chose Decision Trees (DT), Gradient Boosting (GB), Random Forests (RF) and Extreme Gradient Boosting (XGB). For deep learning, we used ConvLSTM, which we will refer to as CL later. It has a classical and effective structure that help improve the feature extraction capability at spatial level compared with traditional LSTM and its variant networks, and is suitable for aggregating temporal and spatial features. ConvLSTM was first used to predict future radar maps on the short-term precipitation problem [32], and was later used in those fields which have spatio-temporal characteristics such as Future Frame Prediction (FFP). In order to evaluate the effect of window self-attention and shifted window attention, we made ablation experiments and added them to the convolution part of ConvLSTM, which is abbreviated as SWA-CL. Similarly, in order to evaluate the effect of Roll-RLNN structure, we also changed the temporal part of ConvLSTM to adapt the 2-D time structure, which we mark it as Roll-CL. UNet [33] is also chosen for comparison as it excels at extracting subtle features that, to some extent, can be used to learn the differences between the forecast results and ground truth. For model parameters, the machine learning-based models are subjected to extensive grid searches for optimal parameter selection. For deep learning-based models, optimization techniques such as residual networks, layer normalization, etc., were also added at the same time to ensure fairness.

In our experiments, our model was implemented by Pytorch, and for the main hyper-parameters, we found that better performance can be expected when we set initial learning rate within a range from $1.5 \times 10^{-3}$ to $4 \times 10^{-3}$, and its decay rate per epoch from 0.9 to 0.96, the minimum learning rate from $1 \times 10^{-4}$ to $5 \times 10^{-4}$, L2 regularization of optimizer (Adam was used in the experiments) from 0 to $5 \times 10^{-2}$, and a small batch of no more than 16, respectively. The small batch may result from the scale of dataset since we only trained it for a period of 6 months. The training loss curve is presented in Figure A1. For SWAR, there is also an important hyper-parameter that we need to concern, which is the sub-window size. For the setting of sub-window size, a reasonable assumption is that we hope a characteristic unit of sea level pressure can be covered as much as possible by a sub-window, so that better features can be extracted in the sub-window, and by a shifted window process the different feature units can get a better information interaction at a higher level. We analyzed the sea level pressure in December 2019, and found that there were many low pressure masses that passed through the area quickly in several days with

their evolutions, which is usually associated with extreme weather phenomenons, such as extreme wind speed, extreme rainstorm, etc., that jointly cause a high forecast error (to some extent) for each forecast institution. Some examples are presented in Figure A2, and we can see that the masses are initiated approximately at a size range from 200 to 400 km, as the latitude and longitude spacing of 1° is approximately equal to 111 km. Therefore, to make these small masses, or the characteristic units be covered by sub-windows, a sub-window size of 4° × 4° is reasonable since it meets the cover requirement and can also be divided with no remainder based on the current condition of data structure. A shifted window process extends the cover size of information, which provides a chance to process the larger pressure masses that are evolved from an initial one. Models based on the sub-window size show effective performance in Section 4.5. However, we used a priori knowledge that states that many masses are initiated approximately at a size range from 200 to 400 km to select the sub-window size. Although it works effectively, it is not the optimal solution for the whole problem. Here we just provided a way to select sub-window size. To theoretically find the optimal method for the whole problem, and the one based on the 2-D time framework, more works are still needed in the future.

### 4.5. Result and Analysis

The results of MME forecasts are shown in Tables 1 and 2, where Tables 1 and 2 show the metrics results with 24–96 h and 120–168 h leading respectively within December 2019. From the experimental results, we can see that the listed non-deep learning models are inferior to the deep learning-based models in these problems, especially after 48 h, which reflects that our assumptions about the characteristics of forecast data are reasonable to some extent. That is, due to the special spatial-temporal structure with 2-D time, traditional machine learning-based methods cannot explicitly specify the means of information transmission and aggregation by designing a network structure like the deep learning-based model. Compared with CL, both Roll-CL and SWA-CL have lower forecast error, indicating that the structure of shifted window attention and Roll-RLNN is effective. For the SWAR that we proposed in this paper, due to the network design for the special spatial-temporal structure in numeric forecast, it achieved significant improvement compared with other models and the best results in almost all the indicators. Although SWA-CL and Roll-CL have a close result to SWAR in the RMSE metric of 24–168 h total, SWAR is far better than the first two in terms of 24 h leading, where the 24 h period is the most critical of all lead times in this paper as it is the most impending and can affect us most quickly. Note that in Section 3.3 we choose the first lead, i.e., when $\beta = 1$, to serve as the basis for the neural network to learn along both time dimensions. Thus, the model performance also shows that our model built a reliable basis for both initial time and lead time, and the integration of the spatial level and the special temporal level can provide more information for the neural network to learn.

**Table 1.** Forecast skill of MME forecasts with 24–96 h leading.

| Model | +24 h | | | +48 h | | | +72 h | | | +96 h | | |
|---|---|---|---|---|---|---|---|---|---|---|---|---|
| | RMSE | MAE | PCC | RMSE | MAE | PCC | RMSE | MAE | PCC | RMSE | MAE | PCC |
| DT | 126.208 | 87.368 | 0.994 | 185.501 | 125.377 | 0.986 | 300.418 | 201.574 | 0.967 | 503.602 | 318.338 | 0.921 |
| GB | 102.731 | 67.270 | 0.996 | 171.424 | 112.343 | 0.988 | 292.453 | 192.398 | 0.969 | 504.759 | 312.386 | 0.919 |
| RF | 94.315 | 63.805 | 0.996 | 167.639 | 110.257 | 0.988 | 292.668 | 192.152 | 0.969 | 503.383 | 311.972 | 0.920 |
| XGB | 92.626 | 67.679 | 0.996 | 163.535 | 109.848 | 0.989 | 283.470 | 187.868 | 0.968 | 469.173 | 293.680 | 0.925 |
| SWA-CL | 89.083 | 65.325 | 0.996 | 151.797 | 105.002 | 0.989 | 264.170 | 178.169 | 0.970 | 445.296 | 282.896 | 0.929 |
| Roll-CL | 91.566 | 67.233 | 0.996 | 153.388 | 106.187 | 0.989 | 262.310 | 176.907 | 0.971 | 440.072 | 280.781 | 0.931 |
| CL | 95.930 | 67.968 | 0.997 | 159.746 | 112.429 | 0.989 | 268.090 | 184.177 | 0.970 | 447.983 | 289.133 | 0.930 |
| UNet | 98.481 | 72.873 | 0.996 | 164.132 | 117.063 | 0.989 | 277.783 | 191.349 | 0.970 | 454.062 | 294.262 | 0.929 |
| SWAR | 83.609 | 59.915 | 0.997 | 150.605 | 103.875 | 0.990 | 261.938 | 176.478 | 0.971 | 439.511 | 279.317 | 0.932 |

**Table 2.** Forecast skill of MME forecasts with 120–168 h leading.

| Model | +120 h | | | +144 h | | | +168 h | | | 24–168 h Total | |
|---|---|---|---|---|---|---|---|---|---|---|---|
| | **RMSE** | **MAE** | **PCC** | **RMSE** | **MAE** | **PCC** | **RMSE** | **MAE** | **PCC** | **RMSE** | **MAE** |
| DT | 642.046 | 430.539 | 0.867 | 717.626 | 517.745 | 0.826 | 783.186 | 590.553 | 0.781 | 525.737 | 324.499 |
| GB | 638.788 | 427.813 | 0.867 | 706.723 | 511.629 | 0.828 | 786.386 | 592.114 | 0.779 | 521.832 | 316.565 |
| RF | 644.399 | 429.011 | 0.866 | 711.593 | 516.490 | 0.826 | 789.610 | 594.304 | 0.776 | 523.879 | 316.856 |
| XGB | 594.146 | 392.576 | 0.885 | 658.684 | 473.069 | 0.854 | 738.228 | 552.677 | 0.808 | 488.095 | 296.771 |
| SWA-CL | 563.566 | 375.682 | 0.889 | 626.763 | 454.223 | 0.861 | 709.431 | 531.911 | 0.822 | 464.878 | 284.744 |
| Roll-CL | 561.720 | 373.089 | 0.889 | 631.069 | 454.889 | 0.860 | 714.089 | 534.695 | 0.821 | 465.692 | 284.826 |
| CL | 567.243 | 377.326 | 0.888 | 631.731 | 454.207 | 0.860 | 720.928 | 535.804 | 0.820 | 470.238 | 288.721 |
| UNet | 576.224 | 382.694 | 0.887 | 633.582 | 455.086 | 0.860 | 706.553 | 530.050 | 0.823 | 470.964 | 291.911 |
| SWAR | 562.694 | 371.838 | 0.889 | 628.396 | 451.946 | 0.861 | 711.579 | 531.609 | 0.821 | 464.344 | 282.140 |

In addition, we further analyze the spatial distribution of the forecast performance of our model, and we use the RMSE change rate as the evaluation. In Figure 6, the experimental results show that SWAR effectively reduces the original forecast errors of all the institutions in most areas, of which the largest reduction is relative to CMA, with most areas reduced by more than 30%.

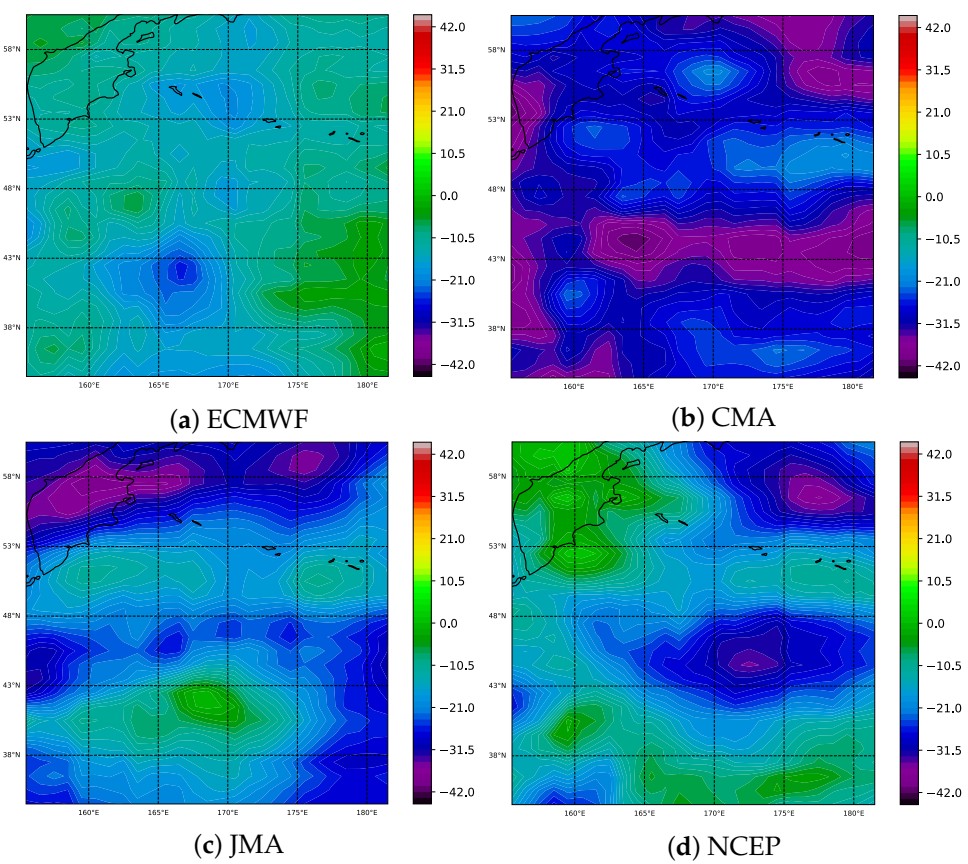

**Figure 6.** (**a**–**d**) show the spatial distribution of the percentage change in RMSE of SWAR compared with ECMWF, CMA, JMA, and NCEP, respectively.

*4.6. Discussion*

The data used for training and validation in this paper mainly includes five dimensions, i.e., initial time, lead time, longitude, latitude, forecast institution or type of numerical model. Our model can make use of information from five dimensions at the same time, and we have verified the performance of the model at temporal and spatial level. From the perspective of output, our model can forecast all the lead times of points in a region at

the same time. This means that in a running server, whenever several new forecasts are produced by their own forecast institution, where each forecast contains a series of output results corresponding to the lead time, our model can make the MME forecast immediately. At the same time, our model does not need additional auxiliary factors and requires only a few ensemble members, which means that it can also be used in the environment where forecast data or experience is insufficient.

## 5. Conclusions

Currently , forecast data with grid and 2-D time structure widely exists in the NWP institutions whose numeric models are run for service. Thus, designing a model that is suitable for the special structure to improve forecast quality is of great significance. In the future, related work might involve evaluating the applicability of more models based on Roll-RLNN for more meteorologic elements, such as temperature, humidity, etc. At the same time, more work could focus on further expanding the dimensionality of the MME forecast model, such as expanding the scalar elements such as sea-level pressure and temperature to vector elements, such as wind, or focusing on expanding single element to multiple elements, which means the model can forecast elements such as pressure, temperature, wind, etc., with multiple lead times and spatial points in a region simultaneously. Finally, it is also possible to expand MME forecast from 2-D space to 3-D where the meteorologic elements at different pressure levels represent the height dimension.

**Author Contributions:** Investigation, B.J.; methodology, J.Z. and L.X.; experiment, J.Z.; validation, B.J. and L.X.; visualization, J.Z.; writing—original draft, J.Z.; writing—review and editing, B.J. and L.X. All authors have read and agreed to the published version of the manuscript.

**Funding:** This research was funded by the National Key Research and Development Program of China grant number 2021YFC3101600.

**Data Availability Statement:** Publicly available data sets were analyzed in this study. The data sets can be obtained from: https://apps.ecmwf.int/datasets/ (accessed on 3 November 2022).

**Conflicts of Interest:** The authors declare no conflict of interest.

## Appendix A

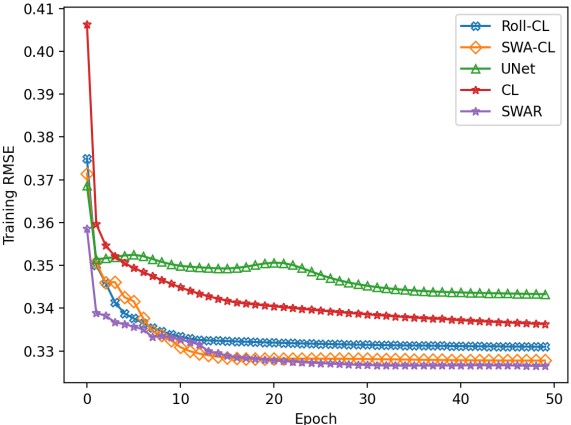

**Figure A1.** Training RMSE of deep learning-based models at a z-score normalization level is shown. The decrease of RMSE curve can be regarded as the adjustment process for the original forecast error.

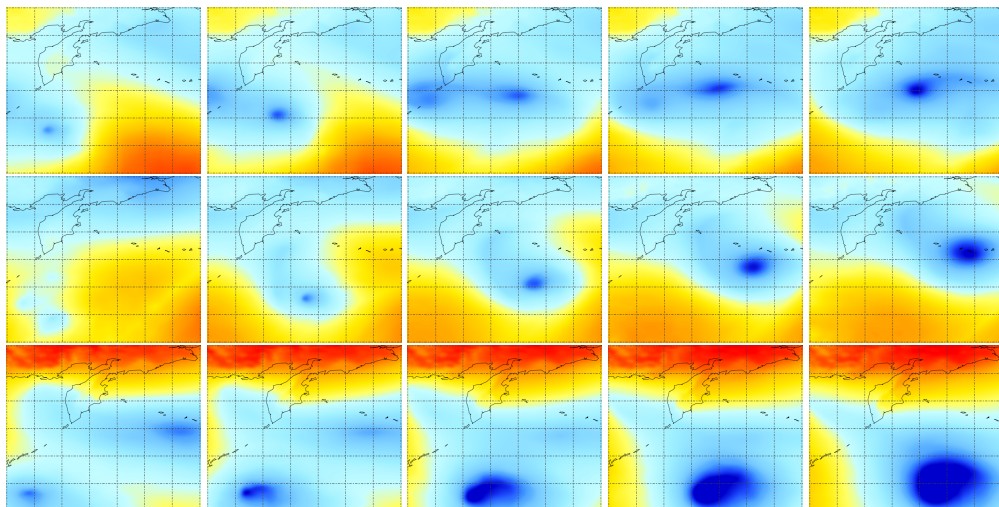

**Figure A2.** Examples of low pressure masses and their evolutions in December 2019 are shown in the above figures. The spatial ranges cover the region from 32.0° N to 59.5° N and 155° E to 177.5° W. The Rows represent the different examples, while columns represent the evolution process. Grid lines in a certain figure have the latitude and longitude spacing of 5°. Red and blue color represent the high and low pressure respectively with the range from 95,955.9 Pa to 104,724.2 Pa.

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
