# Peer review of "SWAR: A Deep Multi-Model Ensemble Forecast Method with Spatial Grid and 2-D Time Structure Adaptability for Sea Level Pressure"

_information, doi:10.3390/info13120577_

Round 1
Reviewer 1 Report
The paper “SWAR: a deep multi-model ensemble forecast method with spatial grid
and 2-D time structure adaptability for sea level pressure” applied shifted window self-attention deep learning method to predict sea level pressure. Their method shows a good performance compared to other methods.
However, I have the following concerns for the authors,
Main Concerns
(1) The method is still relying on choosing a good window size like other meshing methods. The paper lacks discussion about how to choose the best window size. Theoretically, the larger window size is faster but lacks details and in deep learning , larger window sizes will introduce model collapse. For smaller window sizes the computation burden will be heavy. So a discussion about how to choose those sizes are key for the prediction success.
(2) The authors shall provide more details about their model, for example, how to code the attention based deep learning model e.g. is it based on Keras, PyTorch or TensorFlow? In addition, a discussion about how to select hyper-parameter like optimizer, learning rate, epoch number etc would be useful. Attached a figure about the training loss curve would also be useful for the reviewer to better estimate the effectiveness of training.
(3) In Figure 5, could the authors better explain why there is an increasing trend of RMSE as leading time increases? And based on the figure, RMSE seems linearly proportional to leading time. This worries me that there might be model drift as numerical error accumulates. Or this is expected by the authors since it could be explained by other analyses?
(4) Window self-attention is not new, for example, Long Transformer and etc, why the author does not cite any references in section 3.1? They do cite attention mechanisms but ignore citations for window self-attention.
Small Issues
(1) In Page 2 Paragraph 2, GCM is mentioned before showing the full word, shall explain the full word before using the acronym.
Reviewer 2 Report
A deep MME forecast method suiting a special structure has been proposed. At spatial level, the model uses window self-attention and shifted window attention to aggregate information.
At temporal level, they propose a recurrent like neural network with a rolling structure (Roll-RLNN) .
A deep experimental analysis has been conducted for showing that the model structure is effective and can make significant forecast improvements.
Some minor comments/suggestions:
-Some texts in Figure is too small, please adjust
- I did not understand what "oc" is (page 7 and 8)
